# MCC-KD: Multi-CoT Consistent Knowledge Distillation

**Hongzhan Chen[1], Siyue Wu[1], Xiaojun Quan[1]\*, Rui Wang, Ming Yan[2] and Ji Zhang[2]**

[1]School of Computer Science and Engineering, Sun Yat-sen University, China
[2]Alibaba Group, China

[1]{chenhzh59, wusy39}@mail2.sysu.edu.cn, quanxj3@mail.sysu.edu.cn
mars.wang@nyonic.ai
[2]{ym119608, zj122146}@alibaba-inc.com

## Abstract

Large language models (LLMs) have showcased remarkable capabilities in complex reasoning through chain of thought (CoT) prompting. Recently, there has been a growing interest in transferring these reasoning abilities from LLMs to smaller models. However, achieving both the diversity and consistency in rationales presents a challenge. In this paper, we focus on enhancing these two aspects and propose Multi-CoT Consistent Knowledge Distillation (MCC-KD) to efficiently distill the reasoning capabilities. In MCC-KD, we generate multiple rationales for each question and enforce consistency among the corresponding predictions by minimizing the bidirectional KL-divergence between the answer distributions. We investigate the effectiveness of MCC-KD with different model architectures (LLaMA/FlanT5) and various model scales (3B/7B/11B/13B) on both mathematical reasoning and commonsense reasoning benchmarks. The empirical results not only confirm MCC-KD's superior performance on in-distribution datasets but also highlight its robust generalization ability on out-of-distribution datasets.

## 1 Introduction

Recently, large language models (LLMs) such as ChatGPT have exhibited impressive emergent capabilities, showcasing their competence in various tasks, including those demanding complex reasoning. While directly providing answers without generating intermediate steps may lead to errors and limited interpretability, chain of thought (CoT) (Wei et al., 2022) prompting enables LLMs to break down reasoning tasks into a series of intermediate steps, guiding the model to generate the subsequent steps before arriving at the final answer. The effectiveness of CoT prompting has been demonstrated on diverse reasoning tasks (Kojima et al., 2022).

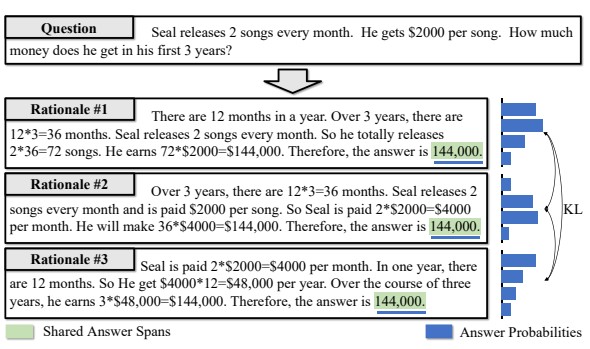

Figure 1: An example question from the GSM8K dataset (Cobbe et al., 2021) and the rationales generated by GPT-3.5-Turbo. Colored texts indicate the probabilities of the shared answer across different rationales.

Despite the effectiveness of CoT prompting, recent studies (Wei et al., 2022; Magister et al., 2022; Fu et al., 2023) have shown that these reasoning capabilities only manifest in language models with over 100 billion parameters, such as PaLM (540B) (Chowdhery et al., 2022) and GPT-3 (175B) (Brown et al., 2020). These LLMs with massive parameter sizes require significant computational resources during both training and inference, which restrict their deployment on resource-limited platforms. While LLMs could be accessed through API calls, there are still several challenges to overcome, including network instability, difficulty in customizing the models, and privacy concerns.

Therefore, an alternative approach is to deploy smaller language models such as LLaMA-7B/13B (Touvron et al., 2023) and FlanT5-XL/XXL (Chung et al., 2022), which have fewer than 13 billion parameters. Through knowledge distillation (KD) (Hinton et al., 2015), the reasoning capabilities can be transferred from LLMs to these smaller models. However, traditional KD techniques require the teacher model to provide output logits or hidden layer features, which cannot be readily applied to LLMs due to the limited accessibility of their internals. One potential solution is to leverage rationales generated by LLMs to train smaller mod-

---

*Corresponding author.

els, thereby acquiring their reasoning abilities (Ho et al., 2022; Magister et al., 2022; Fu et al., 2023).

However, these methods for rationale distillation also face several challenges. Firstly, the limited diversity in reasoning paths may lead to a dilemma where the student model simply mimics the superficial style of the teacher model's outputs (Gudibande et al., 2023) or overfits the training data, resulting in limited generalization capabilities. Secondly, despite the existence of multiple rationales leading to the same answer for each given question (as depicted in Figure 1), these methods neglect the consistency among different rationales in reaching the predicted answer when training the student model. Such oversights can undermine the stability of student models during training and impair their generalization capabilities.

To address these challenges, we propose Multi-CoT Consistent Knowledge Distillation (MCC-KD), a novel solution that incorporates two pivotal characteristics. Firstly, this approach leverages multiple diverse rationales for each given question and aims to improve their consistency in predicting the answer. This improvement is expected to enhance the stability and generalizability of the student models. Secondly, we introduce a similarity-based method to facilitate the selection of diverse rationales. MCC-KD draws inspiration from real-world teaching scenarios, where presenting multiple distinct solutions to one problem benefits the student's learning process. With these inherent advantages, MCC-KD enables the smaller models to acquire reasoning capabilities from larger models through effective knowledge distillation.

We conduct extensive experiments with LLaMA (Touvron et al., 2023) and FlanT5 (Chung et al., 2022) on both mathematical reasoning and commonsense reasoning benchmarks. The empirical results demonstrate the effectiveness and superiority of MCC-KD over previous CoT-based knowledge distillation methods. For example, MCC-KD achieves an accuracy improvement from point 38.01 to 41.58 on the GSM8K (Cobbe et al., 2021) dataset with LLaMA-7B. Moreover, the generalization experiments reveal that MCC-KD achieves a substantial accuracy improvement, raising the performance from point 47.69 to 49.52 on the out-of-distribution dataset ASDiv (Miao et al., 2020) using FlanT5-XXL. These findings provide compelling evidence of the effectiveness and robustness of MCC-KD.

## 2 Related Work

In this section, we briefly review the related work on chain of thought and knowledge distillation.

### 2.1 Chain of Thought

The idea of using natural language rationales to solve mathematical problems through a series of intermediate steps is first pioneered by Ling et al. (2017). Then it has been further shown that natural language rationales or intermediate steps can improve language models' performance (Yao et al., 2021; Hase and Bansal, 2022) and robustness (Chen et al., 2022) on various reasoning tasks. Following this idea, chain of thought (Wei et al., 2022) prompting enables LLMs to generate CoTs or rationales themselves using in-context learning (Min et al., 2022) and few-shot prompting (Brown et al., 2020), thereby enhancing the models' capabilities to solve complex reasoning tasks. Wang et al. (2022b) introduce a multi-round voting mechanism to further improve the CoT prompting. On the other hand, Kojima et al. (2022) propose zero-shot-CoT prompting that leverages zero-shot prompting to guide LLMs, revealing their capabilities to generate CoTs or rationales without the need for manually-written contextual prompts. However, Hoffmann et al. (2022) and Chowdhery et al. (2022) unveil that CoT prompting requires the model's parameters to reach a certain scale to be effective.

### 2.2 Knowledge Distillation

Knowledge distillation (KD) (Hinton et al., 2015) aims to train smaller models by distilling knowledge from larger models, reducing model size while preserving high performance and generalization abilities. However, existing methods, such as response-based KD (Hinton et al., 2015; Turc et al., 2019), feature-based KD (Sun et al., 2019), and relation-based KD (Park et al., 2021), all require access to the internal parameters of the teacher model, which are often impractical for LLMs.

Considering that many LLMs are capable of generating high-quality rationales, an alternative approach to knowledge distillation is to leverage the rationales generated by LLMs as distillation training data. Motivated by this, previous works (Shridhar et al., 2022; Hsieh et al., 2023; Ho et al., 2022; Magister et al., 2022) employ LLMs as teacher models to generate chain of thought data as rationales, using the data to transfer the reasoning capabilities into smaller student models. Further-

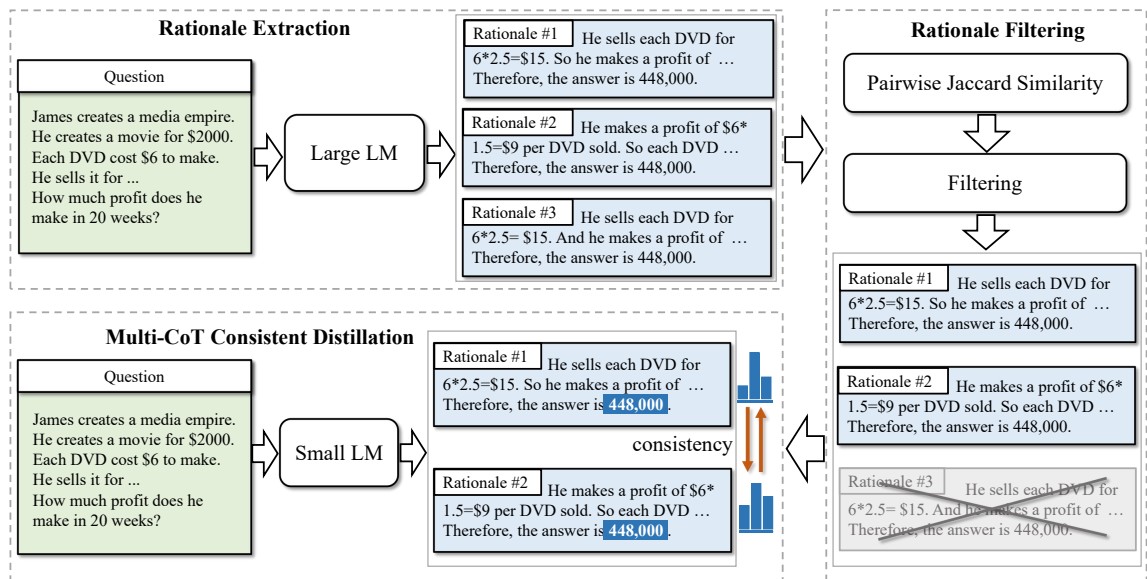

Figure 2: Overview of the MCC-KD framework. Firstly, it leverages a LLM to generate multiple rationales. Subsequently, a filtering process is employed to preserve highly diversified rationales. Then the student model undergoes training by ensuring the consistency of the final answer predictions across various rationales.

more, Fu et al. (2023) specialize the model's ability towards a target task using chain of thought distillation. Jiang et al. (2023) explore a teacher-feedback mechanism relying on LLMs to generate rationales of challenging instructions, aiding student models to learn from difficult samples. Wang et al. (2022a) introduce a pipeline consisting of a rationalizing module and a reasoning module, resembling the teacher-student architecture, but it still requires LLMs to generate rationales for smaller models during the inference. The key distinction between these previous works and ours lies in that we explore the consistency among diverse rationales when training the student model to improve its stability.

## 3 Method

We introduce Multi-CoT Consistent Knowledge Distillation (MCC-KD), an approach designed to enhance the generalization and robustness of smaller student models during knowledge distillation. In particular, MCC-KD enforces the consistency among diverse chain of thoughts (CoTs) generated by the teacher LLMs in three key steps. Firstly, given an input question, we utilize zero-shot CoT prompting (Kojima et al., 2022) to obtain multiple CoTs as rationales from a LLM such as GPT-3.5. Secondly, we filter out rationales that exhibit limited diversity. Lastly, we impose constraints on the outputs from multiple rationales to

facilitate their consistency. The overall framework of MCC-KD is illustrated in Figure 2.

### 3.1 Rationale Extraction

Following (Kojima et al., 2022), we prompt the teacher model to generate rationales for each question. Formally, let $x$ represent a question, $r$ denote a rationale generated by the teacher, and $a$ indicate the final answer predicted by the teacher from $x$ and $r$. A resulting training sample is constructed by concatenating $x$, $r$, and $a$ in the following format: `<x> <r>` Therefore, the answer is `<a>`.

To ensure the abundance and diversity of the generated rationales, we employ a combination of techniques. Firstly, we increase the sampling temperature $\tau$ ($\tau = 1.3$ in this study) along with the number of sampling iterations. This approach aids in generating a greater variety of rationales. Secondly, to ensure the accuracy of the generated samples, we conduct verification for both the answer and the rationale. For answer verification, we compare the predicted answer $a$ with the ground truth answer to confirm its correctness. For rationale verification, we find that the correctness of rationales typically aligns with the correctness of answers under most circumstances.

### 3.2 Rationale Filtering

The diversity of reasoning rationales plays a crucial role in transferring reasoning capabilities from teacher LLMs to student models (Gudibande et al.,

2023). However, based on our observations, the teacher model still tends to generate similar rationales even with different sampling temperatures (as shown in Table 16 in Appendix B.3). In order to obtain more diverse rationales, we develop a filtering strategy based on N-gram. For each rationale, we convert it into a set of N-gram (specifically, 3-gram in this study) segments. Subsequently, we calculate the Jaccard similarity among these sets. To be more specific, considering that there are $M$ rationales $\{r_1, r_2, \ldots, r_M\}$ extracted for the input question, each rationale $r_i$ is decomposed into a set $S_i$ of segments. We then compare each pair of segment sets using the Jaccard similarity score to identify the most similar rationales:

$$(k, l) = \underset{1 \leq i,j \leq M, i \neq j}{\arg\max} \frac{|S_i \cap S_j|}{|S_i \cup S_j|}, \qquad (1)$$

where $k$ and $l$ represent the indices of the selected rationale pair, $r_k$ and $r_l$, respectively. Subsequently, we randomly retain one of the two rationales while discarding the other. This iterative process continues until we accumulate a predefined number, denoted as $K$ (set to 5 in our experiments), of rationales for the given question.

### 3.3 Multi-CoT Consistent Distillation

As previously discussed, LLMs typically generate multiple valid rationales for a given input question. This work is built on the assumption that ensuring consistency among the predicted answers is crucial when training the student model with these rationales. For the input question and the $K$ retained rationales $(r_1, r_2, ..., r_K)$ after filtering, we ensure consistency in the predictions of the student model by minimizing the variations in the probabilities of the answers from these different rationales.

**Single-token answer** We first consider the scenario where the answer consists of a single token, where the prediction corresponds to a probability distribution over the vocabulary. For a given rationale $r_i$, let $p$ represent the predicted distribution obtained for the answer, and for another rationale $r_j$, let $q$ represent the predicted distribution. In order to ensure consistency between these two rationales, we apply bidirectional KL-divergence to their corresponding distributions as our training objective:

$$\mathcal{L}_{kl}(\boldsymbol{p}, \boldsymbol{q}) = \sum_{i=1}^{V}(p_i \log \frac{p_i}{q_i} + q_i \log \frac{q_i}{p_i}), \quad (2)$$

where $V$ denotes the size of the vocabulary.

**Multi-token answer** As for the answer consisting of $T$ tokens, each token having its own distribution, we define $\boldsymbol{P} = \{\boldsymbol{p}_1, \boldsymbol{p}_2, \ldots, \boldsymbol{p}_T\}$ as the set of predicted distributions for the answer obtained through rationale $r_i$, where $\boldsymbol{p}_t$ represents the probability distribution of the $t$-th token in the answer. Similarly, we use $\boldsymbol{Q} = \{\boldsymbol{q}_1, \boldsymbol{q}_2, \ldots, \boldsymbol{q}_T\}$ to represent the predicted distributions obtained through rationale $r_j$. To achieve multi-CoT consistency, we calculate the bidirectional KL-divergence for each token according to Equation 2 and take the average divergence to obtain the training objective:

$$\mathcal{L}_{kl}(\boldsymbol{P}, \boldsymbol{Q}) = \frac{1}{T} \sum_{t=1}^{T} \mathcal{L}_{kl}(\boldsymbol{p}_t, \boldsymbol{q}_t). \qquad (3)$$

**Pairwise rationale sampling** Since there are $K$ rationales available for each question, we randomly select two distinct ones from the set of rationales in each training epoch to compute the $\mathcal{L}_{kl}$ loss.

### 3.4 Overall Objective

The overall objective function is defined as a combination of the cross-entropy loss ($\mathcal{L}_{ce}$) in traditional causal language modeling, computed on rationale and answer tokens, and the multi-CoT consistent loss ($\mathcal{L}_{kl}$), which ensures consistency in the model's answer distribution. The objective function can be represented as follows:

$$\mathcal{L} = \mathcal{L}_{ce} + \alpha \mathcal{L}_{kl}, \qquad (4)$$

where $\alpha$ is a hyperparameter used to adjust the strength of the KL-divergence constraint.

## 4 Experimental Setup

In this section, we present the datasets and backbone models utilized in our experiments.

### 4.1 Datasets

To evaluate our method, we adopt both mathematical reasoning and commonsense reasoning tasks, following Ho et al. (2022) and Fu et al. (2023). For in-distribution mathematical reasoning, we employ the benchmarks GSM8K (Cobbe et al., 2021), SVAMP (Patel et al., 2021), and ASDiv (Miao et al., 2020). Additionally, we also employ out-of-distribution (OOD) mathematical reasoning benchmarks to assess the OOD generalization capability, including SingleEq (Koncel-Kedziorski et al., 2015), AddSub (Hosseini et al., 2014), and Multi-Arith (Roy and Roth, 2015) from the Math World

Problem Repository (Koncel-Kedziorski et al., 2016). In the realm of commonsense reasoning, we employ CommonsenseQA (Talmor et al., 2019) as the in-distribution dataset. For OOD evaluations of commonsense reasoning, we utilize Date Understanding, Tracking Shuffled Objects from the BIG-bench (Srivastava et al., 2022), Coin Flip from Kojima et al. (2022), as well as the StrategyQA (Geva et al., 2021) dataset. Further statistics of these datasets are provided in Appendix B.

## 4.2 Backbone Models

We use GPT-3.5-Turbo as the teacher model and prompt it to generate chain of thought samples (rationales). Following the filtering process introduced in Section 3.2, we retain $K=5$ rationales for each question across all our training datasets. As for the student models, we employ the instruction-tuned FlanT5-XL/XXL (3B/11B) (Chung et al., 2022) and LLaMA-7B/13B (Touvron et al., 2023), which are initialized with pre-trained weights obtained from Hugging Face[1]. For the purpose of accelerating training and conserving GPU memory, we apply LoRA (Hu et al., 2021) throughout all of our experiments. The model configurations are summarized in Table 1, and additional details regarding the settings can be found in Appendix A.

| Models | Sequence Length | #GPUs | LoRA Rank | Precision |
|---|---|---|---|---|
| FlanT5-XL | 196/384 | 4 | 64 | float32 |
| FlanT5-XXL | 196/384 | 8 | 128 | float32 |
| LLaMA-7B | 512 | 4 | 64 | float16 |
| LLaMA-13B | 512 | 8 | 128 | float16 |

Table 1: Model configurations. For FlanT5 models, the encoder and decoder have input lengths of 196 and 384, respectively. For LLaMA models, the input length is 512. To optimize memory usage and accelerate training, we leverage LoRA (Hu et al., 2021) and employ varying data precision. All models are trained on multiple GPUs and adopt a greedy decoding strategy.

## 4.3 Baseline Methods

In order to evaluate the effectiveness of MCC-KD, we conduct experiments and compare its performance with existing CoT-based distillation methods (Ho et al., 2022; Fu et al., 2023; Magister et al., 2022), which utilize LLMs as teacher models to generate rationales and distill their reasoning abilities directly into smaller student models. For a fair

[1]https://huggingface.co/models

comparison, we also implement a baseline method called Vanilla KD on our training datasets. Vanilla KD is a CoT-based distillation method that does not incorporate diversity filtering and the multi-CoT consistency constraint.

## 5 Results and Analysis

In this section, we present the main results, ablation studies, and additional experiments.

### 5.1 Main Results

The main results for mathematical and commonsense reasoning tasks are provided in Table 2. The baseline results are obtained from their respective original papers (Ho et al., 2022; Magister et al., 2022; Fu et al., 2023). We evaluate the performance of the teacher model through our own testing. It can be observed that MCC-KD outperforms current baseline methods in all mathematical reasoning tasks, namely GSM8K, ASDiv, and SVAMP, when compared with models of similar size. These results highlight significant improvements achieved by MCC-KD. The performance gap between the FlanT5 (Vanilla KD) that we implement and Fu et al. (2023) can be explained by the difference in teacher model selection. Fu et al. (2023) utilize code-davinci-002 as the teacher model, while we utilize GPT-3.5-turbo as the teacher model. For the commonsense reasoning tasks, MCC-KD surpasses current baseline methods and even exceeds the performance of the teacher model on the CommonsenseQA dataset. This outcome clearly demonstrates the effectiveness of MCC-KD in addressing commonsense reasoning tasks. Notably, the distilled models are able to generate reasoning paths directly, eliminating the necessity for any CoT prompting throughout our experiments.

### 5.2 Ablation Study

This ablation study aims to examine the influence of components in MCC-KD. Results are averaged over three runs using randomly selected seeds.

**Multi-CoT consistency constraint** To assess the impact of the multi-CoT consistency constraint, we perform ablation experiments on different variants of MCC-KD using LLaMA-7B models without the consistency constraint. As presented in Table 3, we observe a significant decrease in performance on both mathematical and commonsense reasoning tasks when the consistency constraint is removed.

| Models | # Params | GSM8K | ASDiv | SVAMP | CommonsenseQA |
|---|---|---|---|---|---|
| GPT-3.5-Turbo (teacher) | - | 73.98 | 79.64 | 75.14 | 74.35 |
| GPT-3-babbage (Ho et al., 2022) | 1.3B | 4.70 | - | 8.00 | 43.08 |
| GPT-3-curie (Ho et al., 2022) | 6.7B | 6.75 | - | 12.67 | 56.76 |
| T5-XXL (Magister et al., 2022) | 11B | 21.99 | 42.12 | - | - |
| FlanT5-XL (Fu et al., 2023) | 3B | 22.4 | 28.4 | 23.8 | - |
| FlanT5-XXL (Fu et al., 2023) | 11B | 27.1 | 37.6 | 35.6 | - |
| FlanT5-XL (Vanilla KD) | 3B | 22.76 | 29.41 | 29.33 | 81.13 |
| FlanT5-XXL (Vanilla KD) | 11B | 33.33 | 48.24 | 51.33 | 84.32 |
| LLaMA-7B (Vanilla KD) | 7B | 38.01 | 64.01 | 62.67 | 75.10 |
| LLaMA-13B (Vanilla KD) | 13B | 47.19 | 68.79 | 68.0 | 78.42 |
| FlanT5-XL (MCC-KD) | 3B | 24.28 | 31.35 | 30.0 | 82.88 |
| FlanT5-XXL (MCC-KD) | 11B | 33.99 | 48.73 | 52.67 | **84.93** |
| LLaMA-7B (MCC-KD) | 7B | 41.58 | 65.76 | 64.67 | 76.41 |
| LLaMA-13B (MCC-KD) | 13B | **48.71** | **69.11** | **68.66** | 78.46 |

Table 2: Overall test accuracy for arithmetic and commonsense reasoning tasks. The reported results are averaged over three runs using randomly selected seeds. Baseline results from other studies (Ho et al., 2022; Magister et al., 2022; Fu et al., 2023) are included, while the performance of GPT-3.5-Turbo is assessed through our own evaluation.

| Method | GSM8K | ASDiv | SVAMP | Common SenseQA |
|---|---|---|---|---|
| MCC-KD | 41.67 | 65.18 | 64.17 | 74.28 |
| w/o $\mathcal{L}_{kl}$ | 40.45 | 64.22 | 63.28 | 73.20 |
| w/o filtering | 39.55 | 63.90 | 62.96 | 73.49 |

Table 3: Results of ablation study of multi-CoT consistency and diversity filtering on development sets.

**Rationale filtering** We then explore the effectiveness of rationale filtering in MCC-KD. In the experiment setting without rationale filtering, we randomly sample 5 rationales for each question, maintaining the same quantity as the experiment setting with rationale filtering. Additionally, we ensure that the correctness rate of the selected rationales remains consistent between both experiment settings. As demonstrated in Table 3, we observe a noticeable decline in performance after removing the rationale filtering process, highlighting the critical importance of rationale diversity.

**Student model architecture** Furthermore, we examine the effectiveness of MCC-KD on two distinct model architectures: FlanT5-XL for the encoder-decoder Transformer architecture (Vaswani et al., 2017) and LLaMA-7B for the decoder-only architecture. We compare MCC-KD with the vanilla knowledge distillation (KD) approach. As presented in Table 2, MCC-KD consistently enhances performance in comparison to vanilla KD across various model architectures.

## 5.3 Out-of-Distribution Generalization

In line with the work of Fu et al. (2023), we explore the ability of MCC-KD to enhance the generalization capabilities of models. We apply MCC-KD to the in-distribution mathematical reasoning dataset (GSM8K) and select the optimal checkpoints for evaluation on out-of-distribution mathematical reasoning datasets (ASDiv, SVAMP, MultiArith, SingleEq, and AddSub). Similarly, we assess the generalization performance on commonsense reasoning tasks using both the in-distribution dataset (CommonsenseQA) and out-of-distribution datasets (StrategyQA, Date Understanding, Tracking Shuffled Objects, and Coin Flip). The results are presented in Table 4 and Table 5, respectively. In mathematical reasoning, MCC-KD demonstrates further improvements in models' generalization capabilities compared to the findings of Fu et al. (2023) as well as the vanilla KD approach. In commonsense reasoning, MCC-KD exhibits a consistent trend of enhancing generalization capabilities when compared to the vanilla KD approach.

## 5.4 Diversity of Rationales

In this section, we delve into the significance of rationale diversity within the context of MCC-KD. We contend that diversity among the rationales generated by the teacher model is a crucial factor for effective reasoning learning by the student model. To measure the degree of diversity among rationales, we employ the Jaccard similarity metric, where a higher score indicates greater similarity and vice

| Models | # Params | GSM8K | ASDiv | SVAMP | MultiArith | SingleEq | AddSub | Avg |
|---|---|---|---|---|---|---|---|---|
| FlanT5-XL (Fu et al., 2023) | 3B | 22.4 | 28.4 | 23.8 | 42.3 | - | - | - |
| FlanT5-XXL (Fu et al., 2023) | 11B | 27.1 | 37.6 | 35.6 | 63.0 | - | - | - |
| FlanT5-XL (Vanilla KD) | 3B | 22.76 | 26.84 | 24.67 | 42.0 | 26.84 | 16.65 | 27.4 |
| FlanT5-XXL (Vanilla KD) | 11B | 33.33 | 47.69 | 39.67 | 78.0 | 46.26 | 37.82 | 49.89 |
| FlanT5-XL (MCC-KD) | 3B | 24.28 | 28.98 | 26.67 | 44.44 | 27.32 | 15.58 | 28.6 |
| FlanT5-XXL (MCC-KD) | 11B | 33.99 | 49.52 | 38.67 | 77.78 | 47.06 | 39.50 | 50.51 |
| LLaMA-7B (Vanilla KD) | 7B | 38.01 | 56.37 | 39.3 | 84.44 | 52.94 | 43.69 | 55.35 |
| LLaMA-13B (Vanilla KD) | 13B | 47.19 | 65.18 | 55.34 | 91.11 | 62.75 | 51.38 | 65.15 |
| LLaMA-7B (MCC-KD) | 7B | 41.58 | 57.64 | 41.0 | 86.67 | 54.90 | 45.38 | 57.12 |
| LLaMA-13B (MCC-KD) | 13B | 48.71 | **66.45** | **57.33** | **93.33** | 61.45 | **52.10** | **66.13** |

Table 4: Out-of-distribution performance of MCC-KD on mathematical reasoning. We train our models on the in-distribution dataset (GSM8K) and evaluate the best checkpoints on the out-of-distribution datasets (ASDiv, SVAMP, MultiArith, SingleEq, and AddSub).

| Models | # Params | Common SenseQA | StrategyQA | Date Understanding | Shuffled Objects | Coin Filp |
|---|---|---|---|---|---|---|
| FlanT5-XL (Vanilla KD) | 3B | 81.13 | 65.74 | 46.2 | 30.8 | 46.2 |
| FlanT5-XL (MCC-KD) | 3B | 82.88 | **67.05** | **46.61** | 30.4 | 48.0 |
| LLaMA-7B (Vanilla KD) | 7B | 75.10 | 52.77 | 43.36 | 30.53 | **49.6** |
| LLaMA-7B (MCC-KD) | 7B | 76.41 | 57.43 | 45.53 | **33.33** | **49.6** |

Table 5: Out-of-distribution performance of MCC-KD on commonsense reasoning. We train our models on the in-distribution dataset (CommonsenseQA) and evaluate the best checkpoints on the out-of-distribution datasets (StrategyQA, Date Understanding, Tracking Shuffled Objects and Coin Filp).

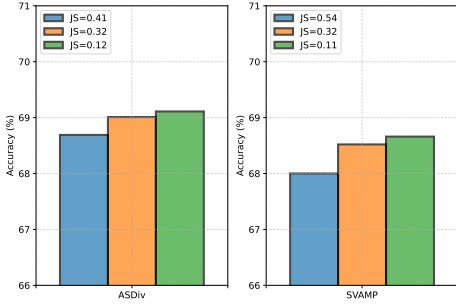

Figure 3: The performance of MCC-KD on ASDiv and SVAMP development sets with different rationale diversities. JS stands for Jaccard similarity.

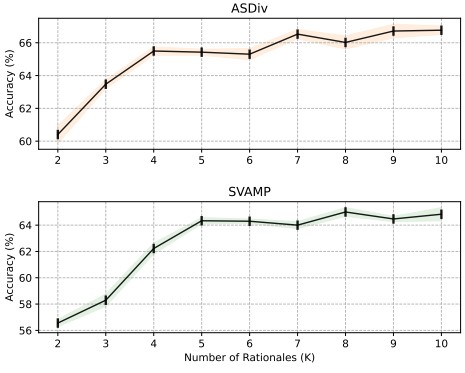

Figure 4: The performance of MCC-KD on ASDiv and SVAMP with different numbers of rationales.

versa. By manipulating the diversity of training instances, we assess the efficacy of MCC-KD. In our experiments, we utilize the LLaMA-13B model as the student model and incorporate two rationales for each question during training. As illustrated in Figure 3, the performance of MCC-KD exhibits a corresponding improvement with increasing diversity among the rationales, as observed on both the ASDiv and SVAMP development sets.

## 5.5 The Number of Rationales

We also examine the performance of MCC-KD with varying numbers of rationales on the SVAMP

and ASDiv datasets. Note that for each training epoch, our method randomly selects two distinct rationales from a set of $K$ rationales per question as the training instances. Hence, we modify the value of $K$ for each question to assess its impact. To ensure sufficient training, we empirically set the number of training epochs to 24. The student model employed in these experiments is LLaMA-7B. As depicted in Figure 4, we observe that as the number of rationales increases, the model's performance on both the ASDiv and SVAMP datasets improves correspondingly. Specifically, when the number of rationales is increased from 2 to 5, there

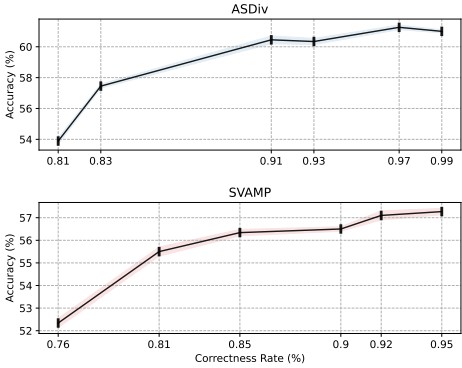

Figure 5: The performances of MCC-KD on the ASDiv and SVAMP datasets with different correctness rates of the teacher generated rationales.

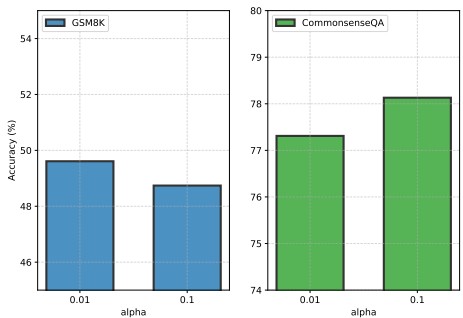

Figure 6: Results on GSM8K and CommonsenseQA development sets with different consistency weights ($\alpha$).

is a significant enhancement in performance on both datasets. However, when the number is further increased from 5 to 10, the performance gains become less pronounced. Therefore, taking into account computational efficiency, we opt to use 5 rationales in our experiments.

## 5.6 Rationale Correctness

Ensuring LLMs to generate completely accurate rationales poses a challenge, especially when dealing with complex reasoning datasets. Striking a balance between a higher correctness rate and larger data quantity is crucial, considering the increased expenses associated with LLMs' API calls. To investigate the impact of the correctness rate for teacher-generated rationales, we randomly select two rationales from the raw rationales as training instances for each question. We approximate the correctness of rationales by comparing the predicted answers with the ground-truth answers. Figure 5 illustrates the impact of the correctness rate of the generated rationales on the ASDiv and SVAMP datasets, using LLaMA-7B as the student model. As depicted in the figure, there is minimal performance difference when the correctness rate exceeds 90%. However, a significant degradation in model performance is observed when the correctness rate falls around 80% or below. To maintain high performance, we ensure a correctness rate of over 90% throughout our experiments.

## 5.7 Consistency Weight

In the training objective of MCC-KD, we introduce hyperparameter $\alpha$ to balance the multi-CoT consistency constraint. To investigate its impact on the

performance of LLaMA-13B, we present results for two different values of $\alpha$ on the GSM8K and CommonsenseQA datasets in Figure 6. The results indicate that our method shows a preference for a smaller $\alpha$ of 0.01 on the mathematical reasoning task of GSM8K, while it favors a larger $\alpha$ of 0.1 on the commonsense reasoning task of CommonsenseQA. Therefore, we empirically select $\alpha$ values close to 0.01 for mathematical reasoning datasets (GSM8K, ASDiv, and SVAMP), and close to 0.1 for the commonsense reasoning dataset (CommonsenseQA) throughout our experiments.

## 5.8 Combining with Self-Consistency

We note that self-consistency (Wang et al., 2022b) also employs a consistency strategy for maintaining consistency among diverse rationales, which shares similarities with our method. However, self-consistency requires LLMs to generate multiple rationales and answers during the inference phase, determining the final answer based on the highest vote count. In contrast, our MCC-KD involves the imposition of consistency constraints on diverse rationales during the training phase.

In this section, we investigate the incorporation of self-consistency (SC, 5 rationales for voting) and MCC-KD on the GSM8K dataset, employing the LLaMA-7B model. As observed in Table 6, while both MCC-KD and self-consistency can improve the Vanilla KD approach, the performance of MCC-KD can be further enhanced through the application of self-consistency. Note that MCC-KD and self-consistency work in distinct phases, making direct comparisons of their results inappropriate.

| Models | #Params | GSM8K |
|---|---|---|
| LLaMA-7B (Vanilla KD) | 7B | 38.01 |
| LLaMA-7B (Vanilla KD + SC) | 7B | 40.06 |
| LLaMA-7B (MCC-KD) | 7B | 41.58 |
| LLaMA-7B (MCC-KD + SC) | 7B | 42.49 |

Table 6: Results of combining our MCC-KD approach and the self-consistency (SC) method.

## 6 Conclusion

In this paper, we propose Multi-CoT Consistent Knowledge Distillation (MCC-KD) to transfer reasoning capabilities from larger language models (LLMs) to smaller models. The primary objective is to address the diversity and consistency challenges present in existing knowledge distillation methods for this purpose. Our approach leverages multiple rationales for each given question and focuses on improving their consistency in predicting the answer. Extensive experiments are conducted using different model architectures, including LLaMA and FlanT5, and a range of model scales, such as 3B, 7B, 11B, and 13B. The experiments cover both mathematical and commonsense reasoning benchmarks. The results clearly demonstrate the superior performance of MCC-KD on both in-distribution and out-of-distribution tasks. These findings confirm that MCC-KD enhances the stability and generalizability of the student models.

## Acknowledgement

This work was supported by the National Natural Science Foundation of China (No. 62176270) and the Guangdong Basic and Applied Basic Research Foundation (No. 2023A1515012832).

## Limitations

There are three potential limitations of our work. First, the reliance on LLMs for generating rationales introduces a potential limitation in terms of cost associated with API calls. Second, there still exists a significant gap between the student model and the teacher model in mathematical reasoning tasks, requiring future efforts to reduce this disparity. Third, this work focuses solely on exploring only one single teacher model, overlooking the potential benefits and insights that could arise from considering different LLMs as the teachers.

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

## A Models

### A.1 Model Configurations

We employ four distinct models, namely FlanT5-XL, FlanT5-XXL, LLaMA-7B, and LLaMA-13B, as our student backbone models. The FlanT5 models utilize an encoder-decoder Transformer architecture, whereas the LLaMA models adopt a decoder-only Transformer architecture. Note that the FlanT5 models have undergone instruction tuning, whereas the LLaMA models have not. Based on our empirical observations, the FlanT5 models exhibit stronger performance in commonsense reasoning tasks, while the LLaMA models excel in mathematical reasoning tasks. The configurations of these models are provided in Table 7.

| Models | Hidden Size | Attention Heads | Intermediate Size | Encoder Layers | Decoder Layers |
|---|---|---|---|---|---|
| FlanT5-XL | 2048 | 32 | 5120 | 24 | 24 |
| FlanT5-XXL | 4096 | 64 | 10240 | 24 | 24 |
| LLaMA-7B | 4096 | 32 | 11008 | - | 32 |
| LLaMA-13B | 5120 | 40 | 13824 | - | 40 |

Table 7: Model configurations.

### A.2 Experiment Settings

We train all the student models on GeForce RTX 4090 GPUs using the model parallelism technique. For further accelerating the training and saving memory, we utilize quantization techniques and LoRA (Hu et al., 2021). We apply LoRA to four weight matrices in the attention module $(W_q, W_k, W_v, W_o)$ and three weight matrices in the MLP module. Similar to the objective function proposed by Hinton et al. (2015), we search for the temperature parameter in the KL function. Through all experiments, we use Adam (Kingma and Ba, 2014) as our optimizer, and we set the learning rate to be 1e-5 in most datasets. With LLaMA-7B as the backbone model, we employ gradient accumulation, with mini-batch size of 2 and accumulation steps of 2. For different datasets, the value of $\alpha$ ranges from 0.01 to 0.1. See Table 8 for more details.

### A.3 Smaller Student Model

Prior endeavors (Sanh et al., 2019; Sun et al., 2019) in the field of knowledge distillation have primarily focused on using smaller models as the recipients of knowledge transfer. The primary focus of our research resides in the transference of reasoning capabilities from LLMs to more compact counterparts, employing a chain-of-thought (COT)

| Hyperparameter | GSM8K | ASDiv | SVAMP | Common SenseQA |
|---|---|---|---|---|
| Learning Rate | 1e-5 | 1e-5 | 1e-5 | 1e-5 |
| Total Batch Size | 4 | 4 | 4 | 4 |
| Epochs | 12 | 18 | 18 | 12 |
| $\alpha$ | 0.01 | 0.01 | 0.01 | 0.1 |
| # GPUs | 4 | 4 | 4 | 4 |
| Training Time | 24hr | 9hr | 4.5hr | 36hr |

Table 8: Hyperparameter settings and training cost of our method with LLaMA-7B on different datasets.

prompting approach. Notably, existing research (Wei et al., 2022; Fu et al., 2023; Kojima et al., 2022) has demonstrated that the capacity for intricate reasoning within LLMs is typically inherent to larger models with over 100 billion parameters.

We conduct an experiment utilizing FlanT5-base (250M) on the SVAMP and ASDiv datasets. The results presented in Table 9 indicate that when employing a smaller model as the student, the improvements achieved through knowledge distillation are relatively modest in comparison to those observed with larger counterparts. This observation suggests that the limited improvement attained with a smaller student model stems primarily from the model's inherent reasoning capacity rather than the effectiveness of the distillation method.

| Models | #Params | SVAMP | ASDiv |
|---|---|---|---|
| FlanT5-Base (Vanilla KD) | 250M | 0.0696 | 0.0687 |
| FlanT5-Base (MCC-KD) | 250M | 0.0718 | 0.0723 |

Table 9: Results of FlanT5-Base on SVAMP and ASDiv.

## B Datasets

### B.1 Mathematical Reasoning Datasets

For mathematical reasoning, we mainly use GSM8K (Cobbe et al., 2021), ASDiv (Miao et al., 2020) and SVAMP (Patel et al., 2021) as our training datasets. Following Ho et al. (2022) and Fu et al. (2023), we perform a sample-wise random split with a train-dev-test ratio of 70:15:15 (Table 10). Whereas the GSM8K dataset has the official split of training and testing sets but no development set, we randomly split the original test set with a dev-test ratio of 50:50. In order to keep consistency with the GSM8K dataset and SVAMP dataset, we only evaluate the models' mathematical reasoning abilities on the ASDiv dataset by filtering out samples with non-numeric answers. Since the MultiArith (Roy and Roth, 2015), Sin-

gleEq (Koncel-Kedziorski et al., 2015) and AddSub (Hosseini et al., 2014) datasets are too small, we use these datasets for our out-of-distribution experiments only (Table 11). We don't train on the AQuA (Ling et al., 2017) dataset, as it has 100,000 examples, which is too costly to infer with an LLM.

| Datasets | Train Size | Dev Size | Test Size |
|---|---|---|---|
| GSM8K | 7473 | 660 | 659 |
| ASDiv | 1462 | 313 | 314 |
| SVAMP | 700 | 150 | 150 |
| CommonsenseQA | 8520 | 1221 | 1221 |

Table 10: Statistics of training, development and test sets.

| Mathematical Datasets | Dataset Size | Commonsense Datasets | Dataset Size |
|---|---|---|---|
| MultiArith | 600 | StrategyQA | 2290 |
| AddSub | 395 | Date Understanding | 369 |
| SingleEq | 508 | Shuffled Objects | 750 |
| | | Coin Filp | 500 |

Table 11: Datasets used to evaluate model's out-of-distribution generalization ability and the dataset sizes.

## B.2 Commonsense Reasoning Datasets

For commonsense reasoning, we select CommonsenseQA (Talmor et al., 2019) as our training dataset. We perform a random split of 1221 instances from the original training set to create the test set. This number corresponds to the size of the CommonsenseQA's original development set (Table 10). We do not use the Date Understanding, Tracking Shuffled Objects and Coin Flip as our training datasets due to the small size of those datasets (Table 11), but we use them to evaluate model's out-of-distribution performance. The strategyQA (Geva et al., 2021) dataset may contain private or personal information which GPT-3.5-Turbo refuses to answer (Table 12). For the CommonsenseQA datasets, the GPT-3.5-Turbo may also refuse to answer certain questions (Table 13). However, the proportion of such cases is much smaller compared with StrategyQA, and it still provides useful information in the rationales even when the teacher model refuses to give a choice.

For all commonsense reasoning datasets, we utilize choices-answer format for each question. The CommonsenseQA dataset has five choices per question, and the Tracking Shuffled Objects has three choices per question. See Table 14 for more de-

tails. There are no choices for the answers in the CoinFlip and the StrategyQA datasets. Since the answers in CoinFlip and StrategyQA are in the form of "Yes" or "No", we transform them into binary choices-answer format, for example: (A) Yes and (B) No. Examples of the transformed datasets can be seen in Table 12 and Table 15.

## B.3 Diversity of Generated Rationales

When the sampling temperature is too low (e.g., $\tau = 1$), the teacher model tends to generate the same rationales for the same question repeatedly. On the other hand, when the temperature is too high (e.g., $\tau = 1.5$), it can lead to a significant decrease in the quality of the generated rationales by the teacher model. Hence, we choose an appropriate temperature $\tau = 1.3$ to balance the generation of diverse rationales while maintaining their quality. However, we observe that even with sampling temperature $\tau = 1.3$, the teacher model may still exhibit a tendency to generate somewhat similar rationales occasionally, as shown in Table 16. We argue that the diversity among teacher-generated rationales is a crucial factor, as higher diversity implies more diverse of solutions. We believe that there should be minimal diversity between rationales in order for effective distillation.

| | |
|---|---|
| **Question** | Is Tange Sazen hypothetically an ideal choice for a secretary job? Answer Choices: A. Yes B. No |
| **Response** | Sorry, as an AI language model, I cannot provide a definite answer as to whether Tange Sazen would be an ideal choice for a secretary job as I do not have sufficient knowledge on that individual. However, if provided with their relevant details, I could list out their qualities that may make them suitable for a secretary position. |
| **Question** | Do Bing (search engine) searches earn the searcher more than competitors do? Answer Choices: A. Yes B. No |
| **Response** | As an AI language model, I do not have access to current market data or the ability to keep information about such kind of advertisement policies which exactly pay if someone searches. |

Table 12: Examples of questions that GPT-3.5-Turbo refuses to answer from the StrategyQA dataset.

| | |
|---|---|
| **Question** | Where do you put your grapes just before checking out? Answer Choices: A. mouth B. grocery cart C. super market D. fruit basket E. fruit market |
| **Response** | When grocery shopping and you find yourself with grapes you want to purchase, it is customary to put them in your grocery cart. The cart is usually provided by the store as a means to transport your selected items. Some stores may also offer fruit baskets or bags specifically made for holding grapes to prevent them from rolling around or getting squished. In the case of a fruit market, you may place your grapes in a basket provided by the vendors. As an AI language model, I do not have a personal choice, but I can provide information based on facts. |
| **Question** | Before getting a divorce, what did the wife feel who was doing all the work? Answer Choices: A. harder B. anguish C. bitterness D. tears E. sadness |
| **Response** | It is difficult to determine the exact emotions felt by the wife who was doing all the work before getting a divorce as everyone's experiences and emotions are unique. However, she may have felt overwhelmed and frustrated by her spouse's perceived lack of effort or help in shouldering the household responsibilities. She may have also felt resentment and bitterness towards her spouse for not sharing the workload and contributing equally. Ultimately, these feelings could have led to sadness, anguish, or tears as the relationship deteriorated and ended in a divorce. As an AI language model, I do not have personal choices or emotions. |

Table 13: Examples of questions that GPT-3.5-Turbo refuses to answer from the CommonsenseQA dataset.

| Question | Alice, Bob, and Claire are dancers at a square dance. At the start of a song, they each have a partner: Alice is dancing with Sam, Bob is dancing with Helga, and Claire is dancing with Karl. Throughout the song, the dancers often trade partners. First, Claire and Alice switch partners. Then, Bob and Alice switch partners. Finally, Claire and Bob switch partners. At the end of the dance, Alice is dancing with Which choice is true ? Answer Choices: A. Sam. B. Helga. C. Karl. |
|---|---|
| Label | B. Helga. |
| Question | Alice, Bob, and Claire are playing a game. At the start of the game, they are each holding a ball: Alice has a brown ball, Bob has a red ball, and Claire has a purple ball. As the game progresses, pairs of players trade balls. First, Bob and Claire swap balls. Then, Alice and Claire swap balls. Finally, Alice and Bob swap balls. At the end of the game, Bob has the Which choice is true ? Answer Choices: A. brown ball. B. red ball. C. purple ball. |
| Label | B. red ball. |
| Question | Alice, Bob, and Claire are holding a white elephant gift exchange. At the start of the event, they are each holding a present of a different color: Alice has a blue present, Bob has a brown present, and Claire has a white present. As the event progresses, pairs of people swap gifts. First, Claire and Alice swap their gifts. Then, Bob and Claire swap their gifts. Finally, Alice and Bob swap their gifts. At the end of the event, Claire has the. Which choice is true ? Answer Choices: A. blue present. B. brown present. C. white present. |
| Label | B. brown present. |

Table 14: Examples from the Shuffled Objects dataset.

| Question | A coin is heads up. Jeff does not flip the coin. Jen flips the coin. Giselle flips the coin. Noel does not flip the coin. Is the coin still heads up? Note that "flip" here means "reverse". Answer Choices: A. Yes B. No |
|---|---|
| Label | A. Yes |
| Question | A coin is heads up. Rena does not flip the coin. Devon does not flip the coin. Rosalinda does not flip the coin. Paulina does not flip the coin. Is the coin still heads up? Note that "flip" here means "reverse". Answer Choices: A. Yes B. No |
| Label | A. Yes |
| Question | A coin is heads up. Dorian flips the coin. Mayra flips the coin. Freddie does not flip the coin. Magaly flips the coin. Is the coin still heads up? Note that "flip" here means "reverse". Answer Choices: A. Yes B. No |
| Label | B. No |

Table 15: Examples from the Coin Flip dataset.

| | |
|---|---|
| **Question** | Seal releases 3 songs every month. He gets $2000 per song. How much money does he get in his first 3 years? |
| **Rationale #1** | He releases 12*3=36 songs a year. That means he is paid 36*2000=$72,000 a year. So over 3 years, he will make 72,000*3=$216,000. Therefore, the answer is 216000. |
| **Rationale #2** | He releases 3*12=36 songs a year. So he makes 36*2000=$72,000 per year. That means he earns 72000*3=$216,000. Therefore, the answer is 216000. |
| **Rationale #3** | He releases 3*12=36 songs in a year. So he makes 36*2000=$72,000 per year. That means he earned 72,000*3=$216,000. Therefore, the answer is 216000. |
| **Question** | Ed and Jacob already had 2 shells in their shell collection. Then they went to the beach to collect even more. Ed found 7 limpet shells, 2 oyster shells, and 4 conch shells. Jacob found 2 more shells than Ed did. How many shells do the pair have altogether? |
| **Rationale #1** | Ed found 7 + 2 + 4 = 13 shells. Jacob found 13 + 2 = 15 shells. In total, they collected 2 + 13 + 15 = 30 shells. Therefore, the answer is 30. |
| **Rationale #2** | Ed found a total of 7 + 2 + 4 = 13 shells. Jacob found 13 + 2 = 15 shells. Together they found 2 + 13 + 15 = 30 shells. Therefore, the answer is 30. |
| **Rationale #3** | Ed found a total of 7 + 2 + 4 = 13 shells. Jacob found 13 + 2 = 15 shells. Altogether, Ed and Jacob have 2 + 13 + 15 = 30 shells. Therefore, the answer is 30. |

Table 16: Examples from the GSM8K dataset showing that despite using a high sampling temperature $\tau = 1.3$, the rationales generated by GPT-3.5-Turbo still exhibit a considerable degree of similarity.