# OpenReview forum: "MCC-KD: Multi-CoT Consistent Knowledge Distillation"
_EMNLP/2023/Conference — EMNLP 2023 Findings_

### Official Review · Reviewer_aS7U · 2023-08-01

**Soundness:** 3

**Excitement:**

3: Ambivalent: It has merits (e.g., it reports state-of-the-art results, the idea is nice), but there are key weaknesses (e.g., it describes incremental work), and it can significantly benefit from another round of revision. However, I won't object to accepting it if my co-reviewers champion it.

**Paper Topic And Main Contributions:**

This paper aims to address the limitations of previous methods of chain-of-thought distillation, including limited diversity of rationales generated from teacher large language models (LLMs) and consistency among different rationales.

These issues can weaken the stability of student models during training and impact their generalization capabilities in a negative way. To rectify these problems, the authors propose a method named Multi-CoT Consistent Knowledge Distillation (MCC-KD).

The proposed method leverages multiple diverse rationales for each question and strives to enhance their consistency in predicting the answer. It uses similarity-based filtering for “diverse rationales” selection. Specifically, the proposed method drops one of the rationales from pairs with high Jaccard similarity, ensuring a diverse set of CoT for each question. By further applying KL divergence on answer generation probability, the method attempts to enforce consistent answer generation across different rationales.

In experiments, the authors validate the proposed method on arithmetic reasoning tasks (GSM8K, ASDiv, SVAMP, …) and commonsense reasoning tasks (CommonsenseQA) and empirically show that the proposed method is better than the vanilla knowledge distillation.

**Questions For The Authors:**

A clear explanation is needed for the model used in Table 3, as its results do not correspond with the performance reported in Table 2.

It would be helpful for the authors to provide insights into why training on CSQA leads to improvement in synthetic tasks like Date understanding, Shuffled objects, and coin flip.

**Reasons To Accept:**

This paper addresses an important problem in chain-of-thought distillation, and the idea of diversifying the generated rationales from LLMs seems plausible. The incorporation of KL divergence regulation to enforce consistency appears to be a sound approach to mitigate the proposed problems in chain-of-thought distillation.

**Reasons To Reject:**

The organization of experimental results is confusing; some of the experimental results seem unrelated to showing the benefits of the proposed methods.
For instance, including results with GPT-3 or T5-XXL as the base LM from other papers in Table 2 does not help readers understand the advantages of the proposed method. The different base LM could lead to varying results, causing unnecessary confusion.
Furthermore, the CommonsenseQA results in Table 2, while showing the superior performance of FlanT5 with MCC-KD, do not provide a comparison with FlanT5 with Vanilla KD.
Table 5 suffers from similar shortcomings which cannot give a comparative evaluation, by providing only LLaMA-13B with MCC-KD results.

In addition, there is a need for a more profound analysis, particularly concerning the idea of filtering with diversity. The authors might consider comparing the proposed method against other filtering methods, such as the one proposed in [1], which is based on reasoning validation. Without such comparisons, especially on hard tasks (e.g., Big-bench Hard[2]) where the teacher model can produce more wrong rationales, this work could potentially mislead future practitioners by leading them to believe that simply diversifying rationales is always beneficial in chain-of-thought distillation.

[1] Ho et al., Large Language Models are Reasoning Teachers, ACL 2023

[2] https://github.com/suzgunmirac/BIG-Bench-Hard

**Reproducibility:**

4: Could mostly reproduce the results, but there may be some variation because of sample variance or minor variations in their interpretation of the protocol or method.

**Reviewer Confidence:**

4: Quite sure. I tried to check the important points carefully. It's unlikely, though conceivable, that I missed something that should affect my ratings.

---

> ### Author Rebuttal · Authors · 2023-08-29
>
> We thank Reviewer aS7U for reviewing our paper and providing helpful feedback on our work.
>
> **Results Organization**
> > Including results with GPT-3 or T5-XXL as the base LM from other papers in Table 2 does not help readers understand the advantages of the proposed method.
>
> Certainly, we agree. The results with GPT-3 or T5-XXL aim to provide a basic comparison. In Table 4, we also incorporated experimental results for FlanT5-XL and LLaMA-7B using the Vanilla KD method. To highlight the benefits of our approach, we are planning to consolidate the data from both Table 2 and Table 4 in the revised version of this paper. Furthermore, we have carried out experiments involving Vanilla KD with FlanT5-XXL and LLaMA-13B, and we plan to include the results of these experiments in the updated Table 2, as outlined below.
>
> | **Models** | **# Params** | **GSM8K** | **ASDiv** |  **SVAMP** |  **CommonsenseQA** |
> | ------- | ------- | ------- | ------- | ------- | ------- |
> | FlanT5-XL (Vanilla KD) | 3B | 22.76 | 29.41 | 29.33 | 81.13 |
> | FlanT5-XXL (Vanilla KD) | 11B  | 33.33 | 48.24 | 51.33 | 84.32 |
> | FlanT5-XL (MCC-KD) | 3B | 24.28 | 31.35 | 30.0 | 82.88 |
> | FlanT5-XXL (MCC-KD) | 11B  | 33.99 | 48.73 | 52.67 | 84.93 |
> | ------- | ------- | ------- | ------- | ------- | ------- |
> | LLaMA-7B (Vanilla KD) | 7B | 38.01 | 64.01 | 62.67 | 75.10 |
> | LLaMA-13B (Vanilla KD) | 13B  | 47.19 | 68.79 | 68.0 | 78.42 |
> | LLaMA-7B (MCC-KD) | 7B | 41.58 | 65.76 | 64.67 | 76.41 |
> | LLaMA-13B (MCC-KD) | 13B  | 48.71 | 69.11 | 68.66 | 78.46 |
>
> **Result Organization**
> > The CommonsenseQA results in Table 2, while showing the superior performance of FlanT5 with MCC-KD, do not provide a comparison with FlanT5 with vanilla-KD.
>
> Indeed, we have conducted experiments using FlanT5-XL and LLaMA-7B with Vanilla KD on CommonsenseQA, as the results show in Table 4. To highlight the advantages of our approach, we will consolidate the results from both Table 2 and Table 4 in the revised version of this paper. Furthermore, we will also include the results of FlanT5-XXL and LLaMA-13B with Vanilla KD in the updated Table 2.
>
> **Result Organization**
> > Table 5 suffers from similar shortcomings which cannot give a comparative evaluation.
>
> Thanks. We will incorporate the results of FlanT5-XL, FlanT5-XXL, LLaMA-13B with Vanilla KD in the updated Table 5 as follows.
>
> | **Models** | **# Params** | **GSM8K** | **ASDiv** |  **SVAMP** |  **MultiArith** |  **SingleEq** |  **AddSub** |
> | ------- | ------- | ------- | ------- | ------- | ------- | ------- | ------- |
> | FlanT5-XL (Vanilla KD) | 3B | 22.76 | 26.84 | 24.67 | 42.0 | 26.84 | 16.65 |
> | FlanT5-XXL (Vanilla KD) | 11B  | 33.33 | 47.69 | 39.67 | 78.0 | 46.26 | 37.82 |
> | FlanT5-XL (MCC-KD) | 3B | 24.28 | 28.98 | 26.67 | 44.44 | 27.32 | 18.58 |
> | FlanT5-XXL (MCC-KD) | 11B  | 33.99 | 49.52 | 38.67 | 77.78 | 47.06 | 39.50 |
> | ------- | ------- | ------- | ------- | ------- | ------- | ------- | ------- |
> | LLaMA-7B (Vanilla KD) | 7B | 38.01 | 56.37 | 39.3 | 84.44 | 52.94 | 43.69 |
> | LLaMA-13B (Vanilla KD) | 13B  | 47.19 | 65.18 | 55.34 | 91.11 | 62.75 | 51.38 |
> | LLaMA-7B (MCC-KD) | 7B | 41.58 | 57.64 | 41.0 | 86.67 | 54.90 | 45.38 |
> | LLaMA-13B (MCC-KD) | 13B  | 48.71 | 66.45 | 57.33 | 93.33 | 61.45 | 52.10 |
>
> **Technical Analysis**
> > There is a need for a more profound analysis, particularly concerning the idea of filtering with diversity. The authors might consider comparing the proposed method against other filtering methods, such as the one proposed in [1], which is based on reasoning validation. Without such comparisons, especially on hard tasks (e.g., Big-bench Hard[2]) where the teacher model can produce more wrong rationales, this work could potentially mislead future practitioners by leading them to believe that simply diversifying rationales is always beneficial in chain-of-thought distillation.
>
> We completely understand the raised concern. We clarify that the concept of diversity filtering is specifically confined to the context of harnessing the capabilities of LLMs for generating reasoning samples and subsequently instructing smaller models through fine-tuning. This is primarily due to the fact that LLMs possess the potential for multiple reasoning paths leading to accurate answers. The effectiveness of reasoning path diversity has indeed been demonstrated in prior studies ([Ho et al., 2022](https://arxiv.org/pdf/2205.11916); [Wang et al., 2023](https://arxiv.org/pdf/2203.11171)).
>
> Furthermore, the filtering process serves another purpose, which is to eliminate samples that yield incorrect answers, ensuring the quality of the training samples. As we continue revising this paper, we do plan to make comparisons with alternative filtering methods. However, we wish to underscore that the core objective of our study isn't solely confined to collecting rationales. Rather, it revolves around the introduction of an innovative knowledge distillation method known as MCC-KD.
>
> **Results Explanation**
> > A clear explanation is needed for the model used in Table 3, as its results do not correspond with the performance reported in Table 2.
>
> As introduced in the captions of Table 2 and Table 3 in the paper, the results with LLaMA-7B in Table 2 correspond to the test set of the arithmetic and commonsense tasks, while the LLaMA-7B results presented in Table 3 correspond to the development sets of the same tasks. We will enhance the clarity of this point in the upcoming revised version.
>
> **Improvement Analysis**
> > It would be helpful for the authors to provide insights into why training on CSQA leads to improvement in synthetic tasks.
>
> The improvement can be attributed to two potential factors. Firstly, the Vanilla KD approach overlooks the consistency among different rationales leading to a predicted answer. In contrast, our MCC-KD method strives to enhance the consistency of these rationales when predicting the answer. This enhances the stability and generalizability of the student models. Secondly, even though the tasks in Table 6 differ, they all share a common format of presenting multiple-choice questions. As a result, our multi-CoT consistent approach derives additional advantages from this structured setup.
>
> Thanks again for your helpful feedback and suggestions.

---

### Official Review · Reviewer_XvaA · 2023-08-02

**Soundness:** 3

**Excitement:**

2: Mediocre: This paper makes marginal contributions (vs non-contemporaneous work), so I would rather not see it in the conference.

**Missing References:**

[1] Automatic Chain of Thought Prompting in Large Language Models. ICLR 2023
[2] Symbolic Chain-of-Thought Distillation: Small Models Can Also "Think" Step-by-Step. ACL 2023 （may be difficult to add this when submitting the manuscript.）

**Paper Topic And Main Contributions:**

This paper proposes Multi-CoT Consistent Knowledge Distillation (MCC-KD) to efficiently distill the reasoning capabilities from ChatGPT to small models such as LLaMA and FLAN.

**Questions For The Authors:**

1. Why don't compare the chain-of-thought reasoning with self-consistency [1]?
2. I understand that improvement by using the math and commonsense datasets, and I also wonder if the performance will drop for other tasks that do not require much reasoning ability.

[1] Self-Consistency Improves Chain of Thought Reasoning in Language Models. ICLR 2023

**Reasons To Accept:**

1. Enhancing the reasoning ability of smaller models is useful.
2. The proposed idea is sound, even though it's not much challenging.

**Reasons To Reject:**

1. Lack of some baselines.
2. The idea of collecting rationales with the most diversity is not novel enough. (see missing refs[1][2])

**Reproducibility:**

4: Could mostly reproduce the results, but there may be some variation because of sample variance or minor variations in their interpretation of the protocol or method.

**Reviewer Confidence:**

4: Quite sure. I tried to check the important points carefully. It's unlikely, though conceivable, that I missed something that should affect my ratings.

---

> ### Author Rebuttal · Authors · 2023-08-29
>
> We thank Reviewer XvaA for providing thoughtful feedback on our work.
>
> **Comparing with Self-Consistency**
> > Lack of some baselines. Why don't compare the chain-of-thought reasoning with self-consistency?
>
> We clarify that a direct comparison between MCC-KD and self-consistency is not feasible. The details are as follows.
>
> - **Self-Consistency Works in Inference** The self-consistency approach entails prompting LLMs to produce multiple rationales and answers during the inference phase. The ultimate answer is then determined by selecting the output with the highest vote count. Consequently, the self-consistency method isn't applicable to the process of knowledge distillation.
>
> - **MCC-KD Works in Training** In contrast, MCC-KD operates by extracting knowledge from GPT-3.5-Turbo, a process that involves the imposition of consistency constraints on diverse rationales during the training phase. This constraint imposition is achieved through the application of bidirectional KL divergence, minimizing disparities between the various rationales.
>
> To sum up, while self-consistency doesn't involve knowledge distillation and relies on a voting-based approach for inference, MCC-KD relies on distilling knowledge from GPT-3.5-Turbo and employs consistency constraints to enhance the consistency among diverse rationales during the training process.
>
> **Technical Novelty**
> > The idea of collecting rationales with the most diversity is not novel enough.
>
> We would like to emphasize that the central objective of this study is not solely centered around the concept of gathering rationales. Instead, our primary focus lies in introducing an innovative knowledge distillation approach named MCC-KD. This approach is designed to elevate the training of student models, enhancing their ability in reasoning tasks. While the technique for collecting diverse rationales holds significance, it constitutes an auxiliary component within the MCC-KD framework. Its role is to ensure the incorporation of a broad spectrum of varied rationales. Similarly, the diversity filtering method proposed, employing Jaccard similarity, serves this very purpose as well.
>
> **Performance on Non-Reasoning Datasets**
> > And I also wonder if the performance will drop for other tasks that do not require much reasoning ability.
>
> We appreciate your question, and we'd like to address it from two perspectives.
> - Firstly, it's crucial to understand that our study's main focus revolves around distilling the reasoning capability of LLMs. Therefore, datasets that don't involve reasoning lie outside the scope of our considerations.
> - Secondly, our approach aims to enhance a student model's capacity for reasoning by ensuring the consistency of diverse rationales leading to the correct answers. Consequently, datasets devoid of reasoning lack such rationales, rendering our approach unsuitable for their application.
>
> Thank you once again for your review! We are more than happy to engage in further discussions.

---

### Official Review · Reviewer_7UzP · 2023-08-05

**Soundness:** 3

**Excitement:**

3: Ambivalent: It has merits (e.g., it reports state-of-the-art results, the idea is nice), but there are key weaknesses (e.g., it describes incremental work), and it can significantly benefit from another round of revision. However, I won't object to accepting it if my co-reviewers champion it.

**Paper Topic And Main Contributions:**

This paper makes a contribution in the space of knowledge distillation (KD) for language models. More specifically, this work focusses on transferring the ability to conduct reasoning through chain-of-thought (cot) prompting from large to smaller models. In order to do this, the authors employ a strategy of generating supporting rationales from large LLMs for a given question, and using a diverse subset of those rationales to conduct KD using a bidirectional KL-divergence loss. The author's demonstrate that this distillation technique leads to superior performance in mathematical reasoning and commonsense reasoning benchmarks.

**Questions For The Authors:**

GPT3-5 turbo is around 154 billion parameters, and the smallest model the authors use for distillation is GPT-3-babbage (around 1.3 billion parameters). Past work in the area of knowledge distillation often focused efforts on much smaller models like DistillBERT (around 66 million parameters), citing the need to deploy language model's in resource constrained computational settings. I wonder if the authors could address why they didn't consider smaller models as candidates for knowledge distillation? Is it because they don't produce a significant increase in performance? If that is the case, it would still be useful to report that as it would help inform the limits of low parameter count models.

**Reasons To Accept:**

The KD strategy proposed in this paper leads to a substantial boost in accuracy for many of the models that were presented in the papers for both mathematical and common sense reasoning. The KD strategy itself takes advantage of pre-trained LLMs to generate rationales and uses and automatable filtering pipeline to diversity the rationale dataset. Therefore, aside from the verification of rationales, this knowledge distillation pipeline can be largely automated and successfully deployed in real-world setting.


**Reasons To Reject:**

Only the rationales of multiple-choice tasks were manually verified. For all other types of rationales the author's write: "For rationale verification, the correctness of the answer typically aligns with the correctness of the rationales in non-multiple- choice tasks". I think a more thorough quantitative analysis is needed to further qualify the validity of the relationship between correct answers and correct rationales. Otherwise, additional manual verification of the prompts is needed especially since they are LLM-generated text.

**Reproducibility:**

4: Could mostly reproduce the results, but there may be some variation because of sample variance or minor variations in their interpretation of the protocol or method.

**Reviewer Confidence:**

3: Pretty sure, but there's a chance I missed something. Although I have a good feel for this area in general, I did not carefully check the paper's details, e.g., the math, experimental design, or novelty.

---

> ### Author Rebuttal · Authors · 2023-08-29
>
> We thank Reviewer 7UzP for positive comments and helpful feedback on our work.
>
> **Quantitative Analysis**
> > I think a more thorough quantitative analysis is needed to further qualify the validity of the relationship between correct answers and correct rationales.
>
> As introduced in the Method section (Lines 217-228), we executed a comprehensive validation procedure to determine the correctness of the generated samples, including both the responses and their accompanying rationales. To further provide a quantitative analysis, we carried out a new investigation to explore the interrelationship between answer accuracy and the validity of rationales within the GSM8K dataset. By carefully analyzing a representative sample of 100 examples, we observed a remarkable alignment, with the accuracy of answers in the sample coinciding with the correctness of rationales at an impressive rate of 99%. We will incorporate this quantitative analysis when revising our paper.
>
> **Using Smaller Model**
> > I wonder if the authors could address why they didn't consider smaller models as candidates for knowledge distillation? Is it because they don't produce a significant increase in performance?
>
> Indeed, prior endeavors in the field of knowledge distillation, such as DistillBERT and TinyBERT, have primarily focused on using smaller models as the recipients of knowledge transfer. However, it is important to acknowledge that the primary focus of our research resides in the transference of reasoning capabilities from LLMs to more compact counterparts, employing a chain-of-thought (COT) prompting approach. Notably, existing research ([Wei et al., 2023](https://arxiv.org/pdf/2201.11903); [Fu et al., 2023](https://arxiv.org/pdf/2301.12726); [Kojima et al., 2023](https://arxiv.org/pdf/2205.11916)) has demonstrated that the capacity for intricate reasoning within LLMs is typically inherent to larger models with over 100 billion parameters. This explanation clarifies why we didn’t consider extremely small models.
>
> To further demonstrate this, we conduct an experiment utilizing the FlanT5-base (250M) model on the SVAMP and ASDiv datasets. The results presented in the following table indicate that when employing a smaller model as the student, the improvements achieved through knowledge distillation are relatively modest in comparison to those observed with larger counterparts. This observation suggests that the limited improvement attained with a smaller student model stem primarily from the model's inherent reasoning capacity rather than the effectiveness of the distillation method itself.
>
> | **Models** | **# Params** | **SVAMP** | **ASDiv** |
> | ------- | ------- | ------- | ------- |
> | FlanT5-Base (Vanilla KD)   | 250M  | 0.0696 | 0.0687 |
> | FlanT5-Base (MCC-KD)   |  250M | 0.0718 | 0.0723 |
>
> Thanks again for your review! Please let us know if you have any further questions, and we are happy to discuss further.

---

### Meta-Review · Area_Chair_Hfbt · 2023-09-19

**Recommendation:** 3

**Metareview:**

The study introduces the novel Multi-CoT Consistent Knowledge Distillation (MCC-KD) method, aimed at the efficient distillation of reasoning capabilities from Large Language Models (LLMs) to smaller ones, addressing the challenge of maintaining both diversity and consistency in rationales. This method innovatively generates multiple rationales for each query and emphasizes consistent predictions by reducing the bidirectional KL-divergence. Through extensive experimentation across various model architectures and scales, the authors have demonstrated the superior performance of MCC-KD, particularly emphasizing its strength in both in-distribution and out-of-distribution datasets.

Pros:

Introduction of the MCC-KD approach, aiming to enhance reasoning capabilities in smaller models through CoT prompting.

Demonstrated superior performance in mathematical and commonsense reasoning tasks, suggesting real-world applicability.

Addresses an imperative issue in chain-of-thought distillation, introducing diversity and consistency in the generated rationales from larger models.

Cons:


Somewhat confusing presentation and organization of experimental results, combined with a lack of profound analysis and comparative evaluations, which could mislead interpretations.

Lack of some baselines (partially addressed in rebuttal).

---

### Decision · Program_Chairs · 2023-10-07

**Decision:**

Accept-Findings

**Comment:**

The study introduces the novel Multi-CoT Consistent Knowledge Distillation (MCC-KD) method, aimed at the efficient distillation of reasoning capabilities from Large Language Models (LLMs) to smaller ones, addressing the challenge of maintaining both diversity and consistency in rationales. This method innovatively generates multiple rationales for each query and emphasizes consistent predictions by reducing the bidirectional KL-divergence. Through extensive experimentation across various model architectures and scales, the authors have demonstrated the superior performance of MCC-KD, particularly emphasizing its strength in both in-distribution and out-of-distribution datasets.

Pros:

Introduction of the MCC-KD approach, aiming to enhance reasoning capabilities in smaller models through CoT prompting.

Demonstrated superior performance in mathematical and commonsense reasoning tasks, suggesting real-world applicability.

Addresses an imperative issue in chain-of-thought distillation, introducing diversity and consistency in the generated rationales from larger models.

Cons:


Somewhat confusing presentation and organization of experimental results, combined with a lack of profound analysis and comparative evaluations, which could mislead interpretations.

Lack of some baselines (partially addressed in rebuttal).